

# Efficient estimation of effective hydraulic properties of stratal undulating surface layer using time-lapse multi-channel GPR

Xicai Pan[1], Stefan Jaumann[1], Jiabao Zhang[1], and Kurt Roth[2,3]

[1]Fengqiu Agro-ecological Experimental Station, State Key Laboratory of Soil and Sustainable Agriculture, Institute of Soil Science, Chinese Academy of Sciences, 71 East Beijing Road, 210008, Nanjing, China

[2]Institute of Environmental Physics, Heidelberg University, Im Neuenheimer Feld 229, 69120, Heidelberg, Germany

[3]Interdisplinary Center for Scientific Computing (IWR), Heidelberg University, Im Neuenheimer Feld 205, 69120 Heidelberg, Germany

*Correspondence to*: Xicai Pan (xicai.pan@issas.ac.cn)

**Abstract.** Multi-scale soil architectures in shallow subsurface are widespread in natural and anthropogenic depositional environments, acquisition of the surface stratal structure and hydrological properties are essential to quantify water cycling. Geophysical methods like Ground-Penetrating Radar (GPR) can provide quantitative information like soil architecture and spatiotemporal soil water content distribution for the shallow layer. Concerning the informative multi-dimensional water flow in the surface layer with an undulating bottom at plot scale, this study assesses the feasibility of efficiently estimating soil hydraulic properties using a few time-lapse multi-channel GPR observations, namely soil water storage and layer thickness of the surface layer at a reclamation land near an old river channel. We show that effective hydraulic properties of the surface layer can be obtained with a small number of time-lapse



GPR measurements during a rainfall event. Additionally, we analyse the effect of some key factors controlling the informative lateral water redistribution on the results of the proposed approach using synthetic simulations.

# 1 Introduction

The exchanges of water and energy fluxes between the land surface and the atmosphere are strongly influenced by geomorphic features like landforms and soils (e.g., Corenblit et al., 2011). Surface morphometric features can be comparatively conveniently obtained from remote sensing at sub-meter scale via satellites or drones. In contrast, the acquisition of subsurface soil features like textural and structural information is not so efficient, although they are essential to quantify soil functions in various

ecosystems (e.g., Rihani et al., 2010). Particularly, soils in fluvial depositional environments exhibit a hierarchical stratal architecture at the scale of meter to hundreds of meters, and modeling subsurface water and mass transport are difficult due to large uncertainties in representation of the hydraulic properties (Huggenberger & Aigner, 1999). Apart from these natural depositional systems, land reclamation also forms a surface layer overlying undulating landforms. The surface layer is usually characterized by a flat

surface and an undulating bottom, and the internal lateral water flow can be not negligible in wetting conditions. Model predictions of water flow and nutrient transport in the surface layer are essential to agriculture. However, it is challenging for the acquisition of such soil architecture information and soil hydraulic properties with the commonly used approaches using point measurements as they are costly and time-consuming.



For in-situ estimation of hydraulic properties, inverse methods are commonly used at multi-scales depending on observed state variables (Vrugt et al., 2008). For one-dimensional (1D) in-situ monitoring profile, soil hydraulic parameters for each layer can be reasonably estimated based on a 1D soil hydrological model and a time series of point observations. For larger scale observations, e.g., from field-scale to catchment scale, a stronger assumption of predominant vertical water flow is implicitly set from the 3D soil hydrological model. Alternatively, an assumption of homogeneous soil column for the shallow measurement depth (< 5 cm) is used for the satellite remote sensing observations (e.g., Mohanty, 2013). Overall, these methods ignore lateral water redistribution and usually suffer from insufficient data information concerning the soil architecture and the soil water dynamics. Yet this information is essential to all inverse parameter estimation methods (Bandara et al., 2013).

Quantitative observations of soil architecture and soil water content became viable during the past decade using geophysical methods. In particular, ground-penetrating radar (GPR) has been used for efficiently imaging of soil water content distribution (e.g., Huisman et al., 2003; Weihermüller, et al., 2007) and together with soil architecture (Gerhards et al., 2008; Bradford, 2008; Pan et al., 2012a; Klenk et al., 2016). In view of informative multi-dimensional water flow containing in the time-lapse spatial observations of GPR measurements, the common problem of insufficient data information from 1D observations is mitigated for inverse modeling of soil water dynamics in the vadose zone soils. This comes at the cost of increasing the complexity of inverse modeling and the need for spatial observations of the soil architecture as well as the soil water content. In addition, since the information of small-scale heterogeneities within the layers cannot be quantified for the GPR measurements using the reflection

approach, the inverse modeling approaches only focus on the stratal soil properties and effective soil water flow at plot scale.

There are at least two popular approaches to use GPR measurements in the inverse estimation of soil hydraulic properties. One approach is the coupled inverse modeling of hydrological processes and GPR

measurements. The model for the simulation of the propagation of the electromagnetic wave in the soils is computationally expensive but necessary for full waveform inversion (e.g., Lambot et al., 2009; Busch et al., 2012; Jadoon et al., 2012) or other evaluation approaches (e.g., Buchner et al., 2012; Jaumann & Roth, 2018). The other approach is directly using the evaluated soil water content and depth from GPR measurements in the inverse hydrological modeling, similar to the above mentioned 1D inversion using

point observations, e.g., TDR measurements. In this study, we propose using pre-evaluated GPR data together with 2D inverse hydrological modeling to estimate soil hydraulic properties of the surface soil layer with underlying undulating structures. We use the optimization procedure described by Jaumann & Roth (2017) to estimate the effective hydraulic properties and demonstrate that this approach allows to efficiently estimate effective hydraulic properties of the surface layer at a reclamation land. Controlling

factors of this approach for practical applications are discussed subsequently.

## 2 Scheme of the proposed hydraulic parameter estimation

### 2.1 Simulation model and setup

The muPhi model (Ippisch et al., 2006) is used for simulating two-dimensional Darcian water flow in a variably-saturated isotropic medium. It employs the Richards equation (Richards, 1931) and the soil

hydraulic functions of van Genuchten (1980) and Mualem (1976) for the soil water characteristic $\theta$(h), usually in terms of the water saturation $\Theta$ (–) and the hydraulic conductivity function $K_w$ (h) as shown in Table 1.

In this work, the water dynamics in a two-layer vadose zone with an undulating interface during a rainfall event is simulated. The model domain is discretized with rectangular grids, where the material for the top layer is characterized by five parameters $p_1 = \{\alpha, n, \theta_r, \theta_s, K_s\}$ as shown in Table 1. Since soil porosity of the top layer is relatively easy to obtain by soil coring, $\theta_s$ is assumed to be known beforehand in the evaluation of the GPR observations of the mean soil water content and reflector depth of the interface (e.g., Pan et. al., 2012a). With this assumption, there are four unknown parameters $p_1 = \{\alpha, n, \theta_r, K_s\}$ for the top layer. Material for the bottom layer is sand and the hydraulic parameters $p_2 = \{\alpha', n', \theta_r', \theta_s', K_s'\}$ are set to the values listed in Table 2 for all studied cases. A Neumann no-flow boundary condition is implemented at the both sides and two types of boundary condition are applied at the upper boundary in consideration of numerical convenience. For the period with strong precipitation events, a Neumann condition is applied for the upper boundary, while a Dirichlet boundary condition is applied for the left period with a time series of outflux. For the bottom boundary, a Dirichlet boundary condition is applied with a fixed water pressure of -0.4 m. Equilibrium state is assumed as initial condition at the lower boundary.

## 2.2 Parameter estimation

Given a time series of multi-channel GPR measurements, the surface soil structure and corresponding soil water dynamics were obtained using the evaluation algorithm proposed by Gerhards et al. (2008). Then,





the effective hydraulic parameters were estimated accordingly. The framework of the 2D inversion procedure is shown in Fig. 1.

The initial hydraulic parameters for each ensemble member were generated based on the Latin-hypercube algorithm. The muPhi model (Ippisch et al., 2006) was used to simulate the spatiotemporal soil water dynamics, and an optimization procedure is implemented. Generally, the optimal soil hydraulic parameters $p = \{\alpha, n, \theta_r, K_s, (\alpha', n', \theta_r', \theta_s', K_s')\}$ were determined with an objective function by minimizing the differences between observed and simulated water storages, $l_{obs}(x, t)$, $l_{mod}(x, p, t)$ at location $x$

$$\chi^2(p) = \frac{1}{2} \sum_t^N \sum_x^M \frac{[I_{obs}(x, t) - I_{mod}(p, x, t)]^2}{\sigma_{obs}}, \tag{1}$$

where $M$ is the number of grid cells in $x$-direction and $N$ is the number of time series of observations. The Levenberg-Marquardt algorithm as implemented in Jaumann and Roth (2017) was used to minimize $\chi^2(p)$. Convergence requires 5 to 40 iterations depending on the existence of structural representation errors in the GPR observations. Depending on the data, two different criteria are used in this study. For the synthetic data without considering structural errors, the "optimal stopping" criterion is applied with a number of iterations up to 40. Whereas for the measured data with structural errors the "early stopping" criterion is applied with only 5 iterations to avoid overfitting to the structural errors. All the inversions were conducted in a cluster via parallel computation. The final estimated parameter set was extracted from the ensemble members at the end.

## 3 Applications

In this section, we apply the approach introduced in Section 2 at a field dataset (Sect. 3.1). Afterward, two controlling factors which can be used to improve the performance of the approach are investigated with synthetic studies (Sect. 3.2).

### 3.1 Field study: Daheigang dataset

The field test site is located near the village of Daheigang at 35 °2′ N, 114º33′ E, Fengqiu County, China. As an aeolian-fluvial depositional environment in the Yellow River floodplain, complex soil architecture is widespread in this area. After land reclamation in the middle of 1980s, aeolian-fluvial landforms are rarely seen except nearby stripped dunes in the woods. The surface soils are dominated by Ochric Aquic

Cambisol and Ustic Sandic Entisol (Li et al., 2007). This provides a suitable test site for the proposed approach.

The GPR data used in this study have been introduced in Pan et al. (2012a) but are shortly described in the following for the convenience of the reader. The GPR survey was conducted using an IDS multichannel GPR system (Ingegneria dei Sisteemi S.p.A., Italy), where two antennas operating at a

15 central frequency of 400 MHz were connected in tandem. The setup was employed using three different antenna separations, S1 = S2 = 0.14 m, S3 = 1.94 m and S4 = 1.66 m. Given the investigated reflector depth around 1.0 m, this setup has an accuracy of 0.05 m in depth (Pan et al., 2012b). Five two-dimensional GPR surveys with two antennas were repeated along prefixed parallel lines with a 1.5 m interval spacing on 22, 23, 25, 27, and 29 May 2011 after a heavy rainfall event which followed ten days

without any rainfalls. These GPR measurements were recorded with a resolution of 0.05 m along the

acquisition line. The measuring wavelet data is the result of stacking 12 scans. Each measurement point

was recorded for a time window of 80 ns discretized in 1024 samples.

The GPR measurements were evaluated using the multi-channel GPR method (Gerhards et al., 2008).

This approach has been used in several studies (e.g. Wollschlager et al., 2010; Westermann et al., 2010;

Pan et. al., 2012a) and uses the travel times from the four channels as common-midpoint (CMP) recorded

during the measurement of each trace along a survey line. To obtain absolute travel times for all channels,

time-zero calibration was applied (Pan et. al., 2012a). Wide-angle reflection-refraction (WARR)

measurements were conducted in air to get the time zero of the cross antenna channels, offsets of the two

box-internal channels were assumed to be equal to the travel time of air-wave wavelet. Thus, reflector

depth $d$ (m) and soil dielectric permittivity number $\varepsilon_r$ (-) can be obtained by inversion. The soil water

content can be derived from a petrophysical relationship between the depth-averaged volumetric water

content $\theta$ (-) and soil dielectric permittivity number (Roth et al., 1990). To mitigate the negative

correlation between $d$ and $\theta$, the total water storage $l = d \cdot \theta$ (m) is used in this study (Pan et. al., 2012b).

### 3.1.1 Setup of the parameter estimation

Data analysis from Pan et al. (2012a) yields an interpolated 3D undulating architecture (Fig. 2) and water

content observations. A qualitative interpretation of the water content difference between the soil above

the trough and hill indicates that lateral flow could happen at such a weather condition (Fig. 3a). To

minimize computational effort, a 2D model of the water dynamics in a transect (magenta section along

the dashed line in Fig. 2) perpendicular to the longitudinal troughs is used that the 2D water flow could

approximate the actual 3D water dynamics. Observations of reflector depth and water storage over the

transect are calculated using the original survey lines (parallel lines at the bottom in Fig. 2) via 3D linear interpolation. Considering the resolution of 0.05 m along the GPR survey lines and a 1.5-m in between the survey lines, the spacing interval of the transect is set 0.1 m. Given a 1-m top layer with a mean soil water content of 15%, the uncertainties of the depth measurement and water storage are set 0.05 m and 0.0075 m, respectively.

For the hydraulic model of the transect, the geometry (16.82 m x 1.3 m) is discretized with a resolution of 0.10 m x 0.05 m (Fig. 3b). The upper Neumann boundary flux (Fig. 3a) is calculated by subtracting evaporation from the natural infiltration flux. The rainfall observations are obtained from rain gauge measurements. The evaporation observations are estimated according to the empirical relationship between reference evaporation and pan evaporation proposed by Allen et al. (1998). Two different empirical coefficients, 0.7 and 0.63 are used for the two consecutive periods, respectively, in view of the removing of land cover of wheat in the morning of 22 May 2011. The lower Dirichlet boundary is constantly set to the position of the water table inferred from drilling (-1.7 m). The initial equilibrated condition is used due to a long precedent dry period. The soil water dynamics in the domain is simulated over a period of 18 days. A number of five time-lapse 2D water storage observations (Fig. 3c) is used for the inverse modeling.

Compared to 1D inverse estimation of hydraulic properties, one major advantage of the proposed 2D approach is using a small number of time-lapse 2D GPR observations which increase the available information about the soil water dynamics. However, apart from the measurement precision, the proposed approach is sensitive to some structural errors presented in the field study. First, the conceptual error can also originate from the simplification of 3D flow to 2D. Second, the GPR derived water storage



observations might contain some structural errors due to the varying accuracy of multi-channel GPR evaluation with different ratios of antenna separation to reflector depth, small-scale soil heterogeneity and the interpolation of unevenly spaced data. Our approach not representing these structural errors may lead to overfitting when the stopping criterion is too small, i.e. $\Delta\chi^2 \leq 1$, or $\chi^2(p) < \chi^2(true)$. Initially, we used the

"optimal stopping" criterion only considering measurement precision and it resulted in a large number of iterations as well as overfitting. After an analysis of the convergence behaviour of the parameter distribution, we identified that overfitting could be avoided by using the "early stopping" criterion and evaluating the output of the 5$^{th}$ iteration.

To increase our understanding of the performance of the proposed approach in the field study, two

additional synthetic studies were setup. First, since the model is forced at the surface, the inversion is more sensitive to the parameters of the first layer than to the parameters of the underlying layer. Thus, an initial synthetic study was conducted to investigate the effect of the parameters of the bottom layer on that of the first layer. For this inversion we used the same settings as the field study but adding unknown hydraulic parameters for the bottom layer. The required synthetic observations of soil water storage were

generated using the forward simulations at the same time as field measurements and adding Gaussian distributed random errors of GPR observations. Hence, the above mentioned structural errors were not included in the inversion and the "optimal stopping" criterion was used. Then, another synthetic study was conducted to investigate the effect of the above mentioned structural errors on the parameter estimation of the first layer. All the settings are the same as the previous synthetic study but using fixed

parameters for the bottom layer.



### 3.1.2 Results and discussion

In the previously described initial synthetic study (Sect. 3.1.1), the parameters for both layers are estimated based on the synthetic data of the first layer. The resulting parameters for the first layer match the true parameter well, but the resulting parameter set for the bottom layer is biased dramatically. Here, we used a measure $S_j = \frac{\partial \chi^2}{\partial p_j}$ to analyze the sensitivity of the inversion to the specific parameters and found that the sensitivity of the parameters of the first layer typically surpasses that of the bottom layer by two orders of magnitude. Therefore, we decided to reduce the computation cost by fixing the parameter of the bottom layer, although this introduces a small structural error. Hereafter, all the inversions only estimate the parameters for the first layer and use the fixed parameters listed in Table 2.

Using the setup described in section 3.1.1, the estimated hydraulic properties for the upper layer from the field study and the synthetic study are shown in Fig. 4. The upper panel Fig. 4a and 4b present the resulting water retention curves and hydraulic conductivity curves, respectively, using the estimated parameters from the field study based on the "early stropping" criterion. We show those 68 best ensemble members with minimum $\chi^2$ accounting for 68% of the 100 ensemble members. Given the observed water content range (18% - 27%, histogram in Fig. 4a), the water retention curve is mainly constraint over a small water content range, and large uncertainties show at both ends of soil water retention curve, in particular at the dry end. Since we set the initial parameters with the Latin Hypercube approach, the parameter space is equidistantly sampled. Due to the low sensitivity of the hydraulic conductivity and the limited iterations of the "early stopping" criterion, the uncertainty of the resulting hydraulic conductivity

is high. The overall uncertainty of the resulting hydraulic conductivity function is relatively high due to the small water dynamic range and measurement errors.

Apart from the influence of the water dynamic range, the ignored structure errors also play an important role in the inversion for the field study. As shown in Fig. 5, the different structures of the standardized residuals in the field study and in the synthetic study indicate that notable structural errors exist in the field study. Once excluding structural errors in the synthetic study, the resulting hydraulic properties in Figure 4c and 4d are much better than the field study. The bands of the resulting hydraulic curves based on the "optimal stopping" criterion are much more narrow than that based on the "early stropping" criterion. The structural errors lead to different histograms in Fig. 4a and 4c, and result in a little bit larger uncertainties of the resulting hydraulic parameters in comparison to the synthetic study when using the same criteria. This is also the reason why overfitting occurs when using the "optimal stopping" criterion in the field study.

In summary, the just passable performance of the proposed approach in the field study is not only attributed to the narrow observed water dynamic range but also the structural errors. However, better performance can be achieved by improving the experiment design. One solution is to deploy the GPR transect measurements directly over the undulating structure with steeper slopes to result in a wider water content range by internal water redistribution. Another solution is to conduct the GPR measurements to capture a wider water content range, i.e. record data before a rainfall event. The two controlling factors are further verified in the following synthetic study.

### 3.2 Synthetic study: Investigation of controlling factors

### 3.2.1 Setup of the parameter estimation

In this section we used synthetic studies to verify the proposed approach and assess the effects of the
slope gradient on the accuracy of the approach. Three two-layer architectures (S1, S2, and S3 in Fig. 6)
are employed with a domain of 6.28 m x 2 m and are discretized with a resolution of 0.04 m x 0.02 m.
The internal layer interface is given as a sine-wave shape with different slope gradients (amplitude: 0.25
m, 0.5 m, 0.75 m), and soil hydraulic properties of both layers are the same as the synthetic case discussed
in Sect. 3.1. Given the same boundary conditions and initial station, parameter estimations are conducted
using the "optimal stopping" criterion as the above study. In addition, the impact of the water dynamic
range on the parameter estimation is also investigated by adding one more measurement of GPR
observations before the heavy rainfall event (Fig. 3a).

### 3.2.2 Results and discussion

The hydraulic functions estimated from the three undulating architectures are shown in Fig. 7. The left
panel compares the estimated water retention curves of S1, S2, and S3 and the right panel compares the
estimated hydraulic conductivity curves. The curves in each plot represent the best 20 estimates with
minimal $\chi 2$, accounting for 67% of the 30 ensemble members. Results from the panels show the
uncertainty band of the estimates decrease sequentially from S1 to S3. In other words, the higher the slope
gradient, the better the estimation. The histograms in Fig. 7a, 7c, and 7e show an increase of water content
range from top to bottom, where the ranges are 19% to 25%, 19% to 26%, and 18% to 30%, respectively.

This is mainly ascribed to the increasing intensity of lateral water redistribution, which results in a wider water dynamic range.

Figure 8 shows the results from the estimates using the same setup as Fig. 7 but extending the observed water dynamic range by adding one more GPR observation. Compared to Fig. 7, performances of the approach in the three architectures are all improved. This is attributed to the extended water content range, which controls on the performance of the inverse estimation. The histograms in Fig. 8a, 8c, and 8e present two peaks and much wider water dynamic range in comparison to Fig. 7, and the ranges all increased to 13% to 26%, 13% to 26%, and 11% to 31%, respectively. However, the improvements of the three architectures are distinct. The most significant improvement happened in the architecture S2 with the intermediate undulating amplitude. The improvement in S1 is not as much as S2, although the soil water content range in S1 is similar to S2. This is attributed to the non-negligible slope-induced lateral water redistribution in S2. Since the inversion is not only influenced by the water content range but also the structural information change resulting from the slope-induced lateral water redistribution. The bell-shaped peaks in Fig. 7a and Fig. 8a indicate little structural information in S1. As a result, only the extended water content range takes a little effect in S1. In contrast, both factors take effect on the inverse estimation for S2 and S3, although the additive effect is not so significant for S3.

Results of both synthetic studies in Fig. 7 and Fig. 8 show that excellent performance of the approach can be achieved when the information of the water content range and slope-induced lateral water redistribution is enough to constrain the inversion. Particularly, the structural information resulting from the slope-induced lateral water redistribution is essential to the proposed approach. To improve the



performance of the approach, other factors contributing to lateral flow such as the soil hydraulic properties and the intensity and duration of the precipitation should be considered in designing field experiments.

## 4. Discussion

For practical application, we remind some necessary conditions for this approach to work. First of all, a surface layer with undulating bottom is required that leads to notable lateral water redistribution. Following this concept, applying this approach for a three-dimensional architecture could be possible but needs further computational and experimental efforts, however. Secondly, for rain-based cases, prerequisite conditions like large rainfall intensity and high soil permeability are necessary to ensure a proper water dynamic range for GPR data acquisition. But the wetting range of soils under natural conditions typically regulated by precipitation controls the applicability of inverse modeling approach (e.g., Steenpass et al., 2011; Scharnagl et al., 2011). Finally, to capture the lateral water redistribution, timing for the time-lapse GPR observations is essential.

Except for the measurement uncertainties in reflector depth and water content, other errors which were ignored in this study influence the application of this method. Three types of errors are summarized as follows. 1) The uncertainty in upper boundary fluxes should be considered when relevant observations are not well conducted. 2) The structural errors in GPR observations from the multi-channel GPR evaluation and the selection of the investigated transect can be reduced with better GPR setup and survey design. 3) Serious conceptual errors might rise up when ignoring small-scale heterogeneous soil properties within the layer. As pointed out in the introduction, the proposed approach focuses on the

stratal soil properties and effective soil water flow at the scale of meter to tens of meters. The layer with strong small-scale heterogeneities is adverse for this approach.

## 5 Conclusions

The surface layer with a flat surface and an undulating bottom at the scale of meter to hundreds of meters
is widespread in natural and anthropogenic affected fluvial depositional environments. This study demonstrates an approach to efficiently estimate effective stratal hydraulic properties of the surface layer. It employs multi-channel GPR to capture soil architecture and time-lapse soil water content, a 2D simulation of soil hydrology, as well as an optimization procedure. The approach is applied at a reclamation land with a surface layer over buried undulating landforms. A small number of time-lapse
GPR observations was conducted after a heavy rain event and effective hydraulic parameters were obtained using 2D inverse modeling. The performance of the proposed approach for the given field data is mainly limited by the observed narrow water content range and structural errors. Further synthetic studies show that better performance can be achieved e.g., by adding one more GPR observation to extend the water dynamic range.
Application of the demonstrated approach mainly relies on the slope gradient of the undulating structure and the lateral water redistribution. Other factors contributing to lateral flow such as soil hydraulic properties as well as the intensity and duration of the precipitation also influence the performance of this approach. Overall, the major advantages of this approach include (i) non-destructive observations, (ii) a bigger scale of the effective soil hydraulic properties, and (iii) efficiency in field applications.





## Acknowledgements

This work is financially supported by the NSFC (National Natural Science Foundation of China) Project

(grant 41771262).



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





**Table 1. Summary of the equations used in the muPhi model (Ippisch et al., 2006) for simulating two-dimensional Darcian water flow in a variably-saturated isotropic medium.**

Governing equation:

1. $\partial_t \theta(h) - \nabla \cdot [K_w(\theta(h))[\nabla h - 1]] = 0$

Hydraulic functions:

2. $\Theta(h) = \dfrac{\theta(h) - \theta_r}{\theta_s - \theta_r} = (1 + |\alpha h|^n)^{-m}$

3. $K(\theta) = K_s \Theta^\tau [1 - (1 - \Theta^{n/(n-1)})^m]^2$

$\theta$, $\theta_s$ and $\theta_r$: the unsaturated, saturated and residual soil water content, respectively, ($m^3 m^{-3}$)

$h$: the hydraulic head (m)

$K_w$ and $K_s$: the unsaturated and saturated hydraulic conductivity, respectively, ($ms^{-1}$)

$\alpha$, $n$, and $m$ ($m = 1-1/n$): empirical parameters shaping the retention curve

$\tau$: the empirical parameter for shaping the hydraulic conductivity function ($\tau=0.5$)



**Table 2. Hydraulic parameters of the two-layer soil and their allowed ranges for parameter estimation. The true values for the upper layer is obtained from Rosetta (Schaap et al., 2001) based on measured soil texture information and the values for the lower layer are cited from Carsel and Parrish (1988).**

|  | Symbol | Description (unit) | True value | Allowed range |
|---|---|---|---|---|
| Upper layer | $\alpha$ | inverse of air entry suction (m$^{-1}$) | -3.91 | -30...-0.1 |
|  | $n$ | measure of the pore-size distribution (-) | 1.7778 | 1.1...10.0 |
|  | $\theta_r$ | residual water content (m$^3$ m$^{-3}$) | 0.0 | 0.0...0.15 |
|  | $\theta_s$ | residual water content (m$^3$ m$^{-3}$) | 0.43 | 0.43 (known) |
|  | $\log_{10}(K_s)$ | $K_s$: saturated hydraulic conductivity (m s$^{-1}$) | -4 | -7...-3 |
| Lower layer | $a'$ | inverse of air entry suction (m$^{-1}$) | -14.5 | -30...-0.1 |
|  | $n'$ | measure of the pore-size distribution (-) | 2.68 | 1.1...10.0 |
|  | $\theta'_r$ | residual water content (m$^3$ m$^{-3}$) | 0.045 | 0.0...0.15 |
|  | $\theta'_s$ | saturated water content (m$^3$ m$^{-3}$) | 0.43 | 0.3...0.5 |
|  | $\log_{10}(K'_s)$ | $K'_s$: saturated hydraulic conductivity (m s$^{-1}$) | -4.08 | -7...-3 |


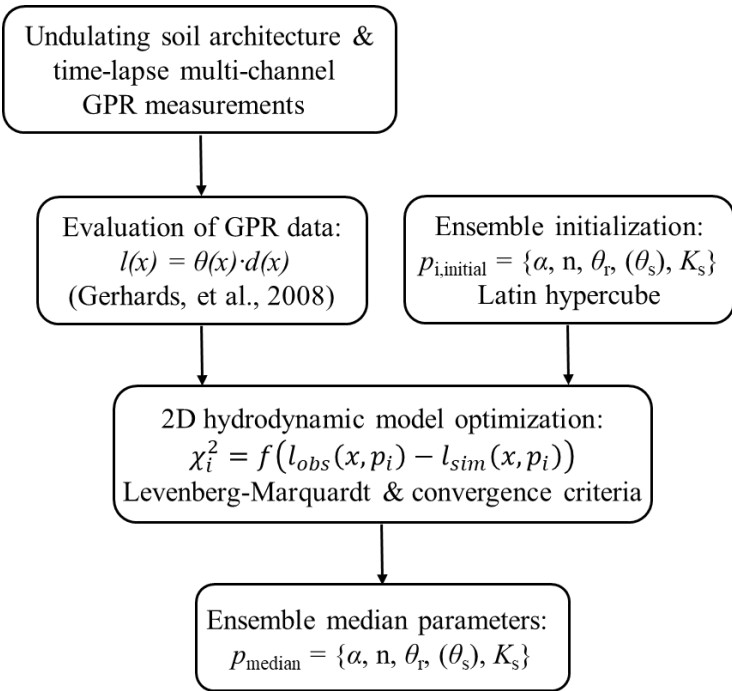

**Figure 1. The framework of efficient estimation of effective hydraulic properties of the surface layer with an undulating interface in a two-layer transect. Provided the water dynamic range captured by a small number of time-lapse multi-channel GPR observations, final parameter sets are inversely estimated based on the Levenberg-Marquardt algorithm and proper convergence criteria (see Sect. 2). Latin-hypercube-sampled initial parameter sets for each ensemble member are used in the 2D inversion. The final median parameter set is derived from the 68% estimated ensemble parameter sets with minimal $\chi^2$.**



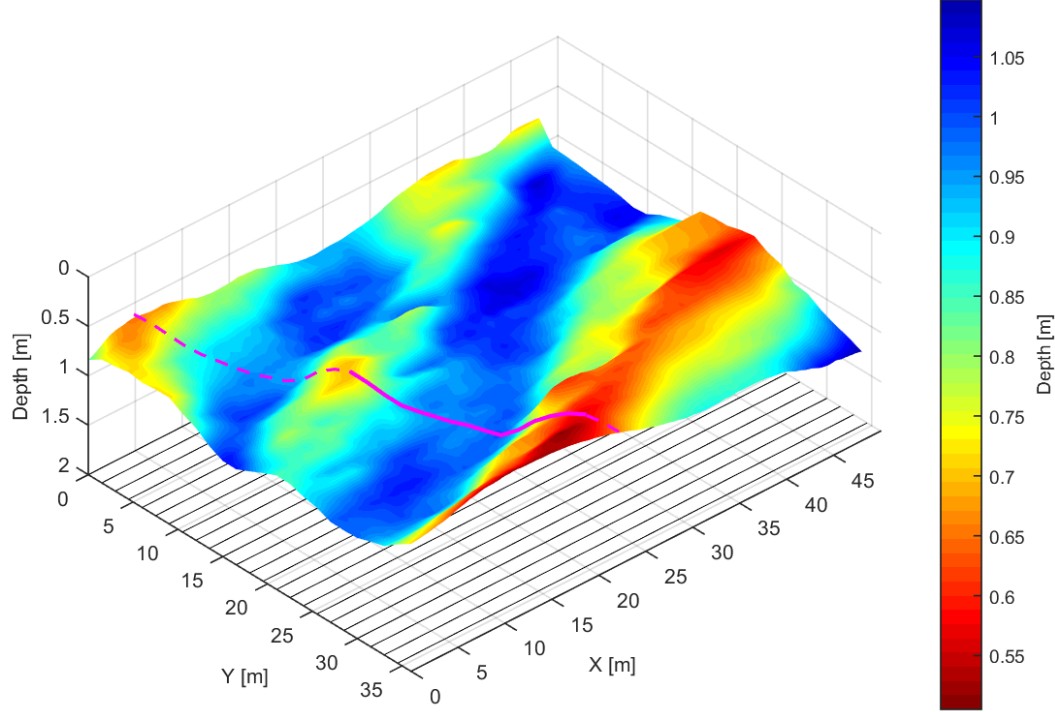

**Figure 2. The underlying interface of soil architecture derived from GPR measurements (adapted from Pan et al. (2012))**

5  **and the selected transect for (magenta curve along the dashed cutting line) perpendicular to the longitudinal troughs.**

**The bottom lines are a projection of the 25 GPR survey lines.**




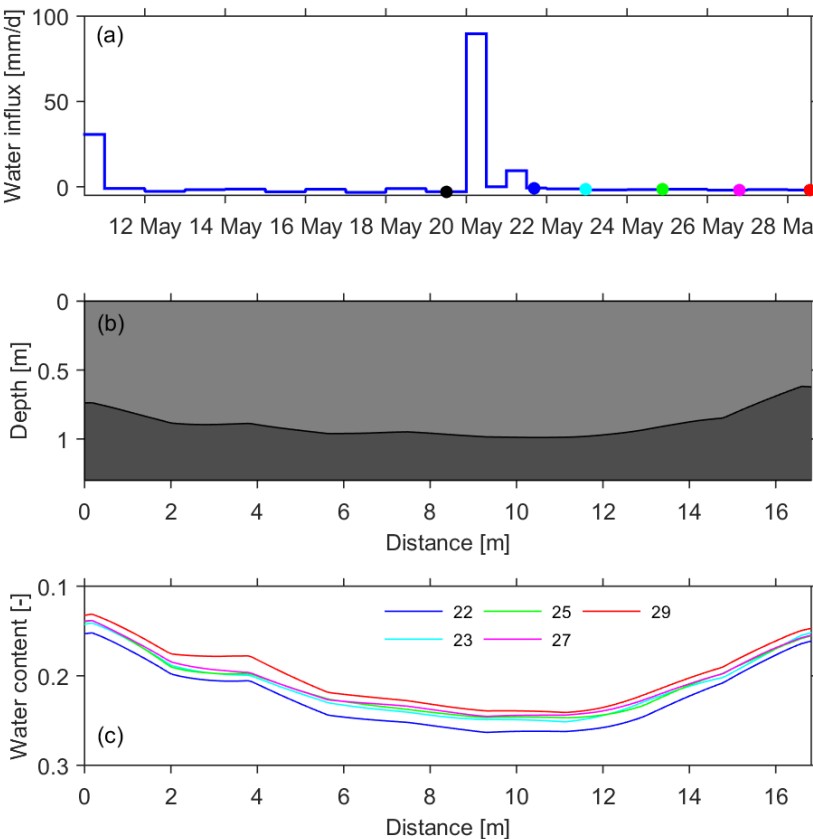

**Figure 3. Setup for the inverse modeling. (a) Observed water flux through the upper boundary. The coloured dots (excluding the black one) correspond to the timing of soil water storage observations in (c). (b) The soil architecture of the transect. (c) Mean soil water content of the upper layer from time-lapse GPR observations on May 22, 23, 25, 27, 29.**



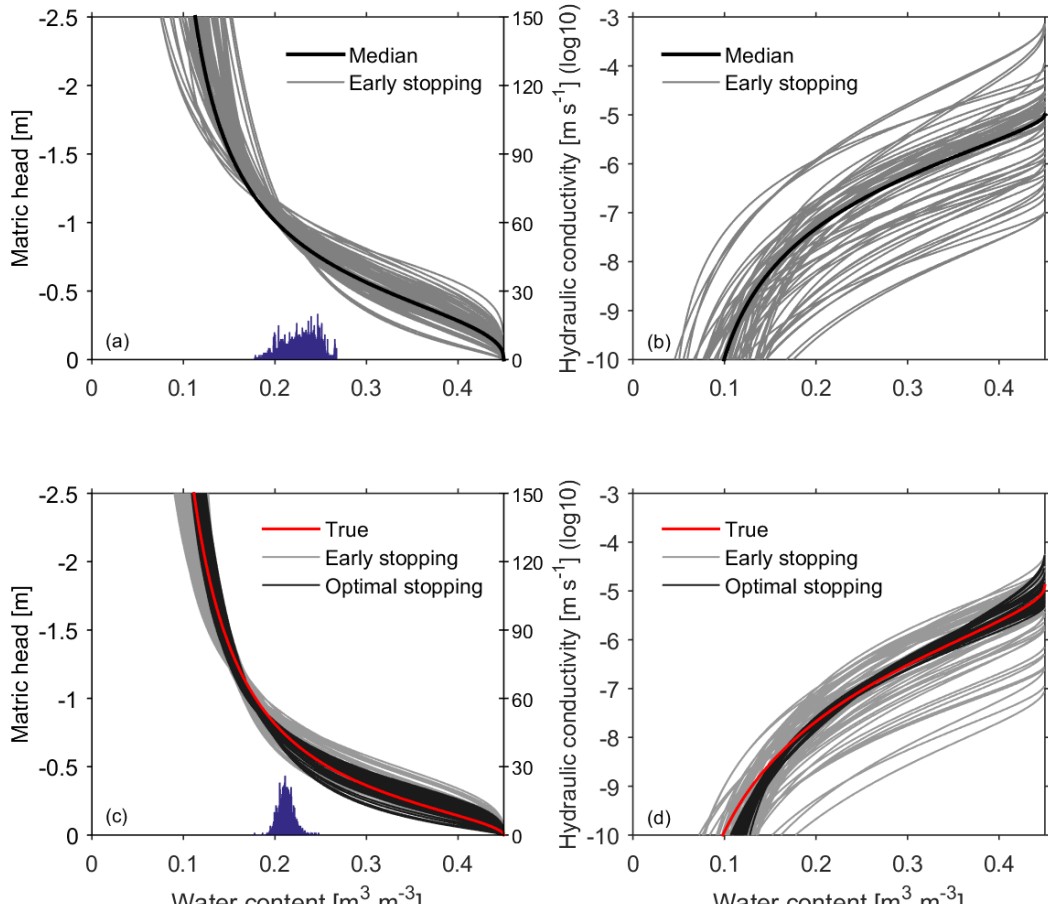

**Figure 4. Effective hydraulic properties estimated with the proposed approach for the field study (upper panel) and the synthetic study (lower panel). Considering that the field study contains the structural errors in the measured data, apart from measurement precision, the "early stopping" criterion (stop with 5 iterations is used to avoid overfitting), while the "optimal stopping" (stop when meeting the convergence criterion) is used in the synthetic study. The "optimal stopping" criterion should be applied once the structural errors are alleviated in the field study. The grey curves in all subplots are estimated hydraulic functions from ensemble members using the "early stopping" criterion. The black curves in (a) and (b) are derived from the median parameters of the best 68 members, while the black curves in (c) and (d) are the same members but using the "optimal stopping" criterion. The red curves in (c) and (d) are derived from the true parameter given in Table 2. The histograms in (a) and (c) (blue histogram, the same in Fig. 6 and Fig. 7) mark the covering range of the mean water content evaluated from the GPR observations.**



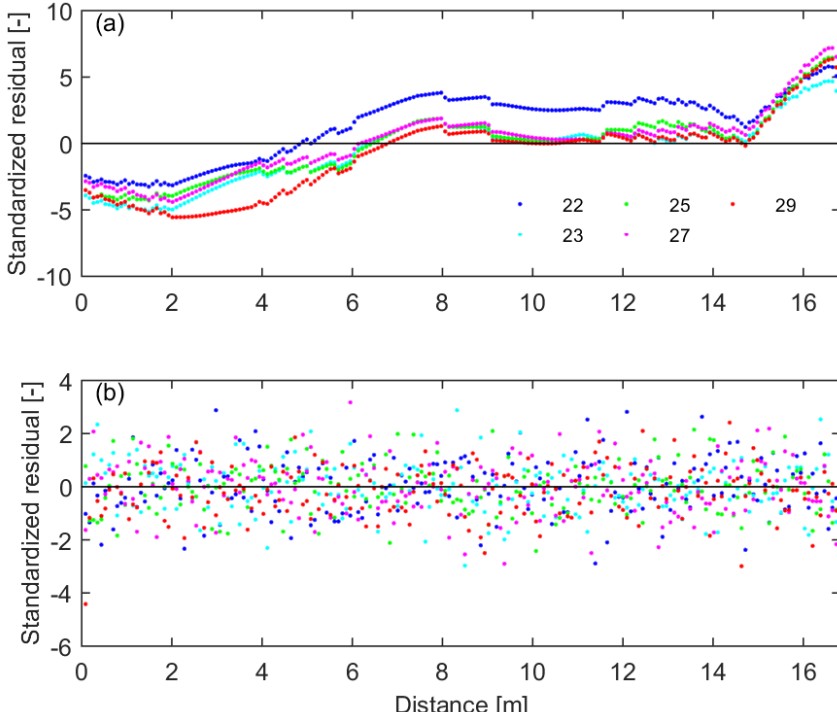

**Figure 5. Impacts of errors on the inverse estimation in the field study and in the synthetic study. (a) structural pattern of standardized residuals in the field study indicates notable structural errors, apart from Gaussian distributed error at each point. (b) Gaussian distributed standardized residuals in the synthetic study.**





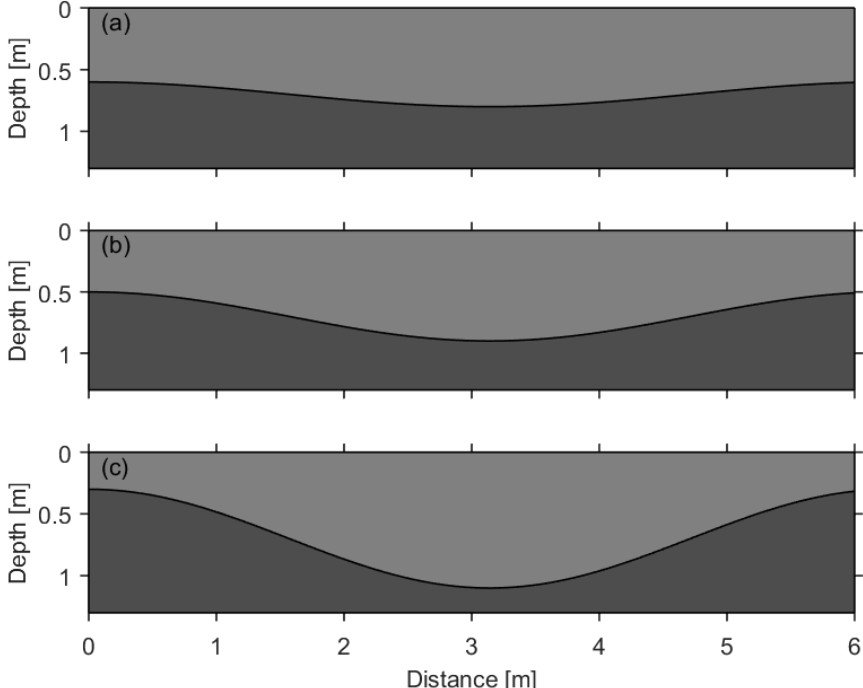

**Figure 6. Undulating soil architectures used in synthetic studies for investigating the performance of the proposed approach with different slope gradients. Three interfaces with different amplitudes are (a) S1: 0.25 m; (b) S2: 0.5 m, and (c) S3: 0.75 m, respectively.**





**Figure 7. Effects of the slope gradient on the performance of the proposed approach. The top, middle and bottom panels compare the results for the soil architectures in Fig. 6. Similar to the lower panel of Fig.4 for the synthetic study, the true parameter set is shown together with the 20 best ensemble members and the median of theses parameter sets. The uncertainty of the resulting hydraulic properties decreases along with the sequentially increase slope gradient and water content range of the histograms from S1 to S3.**





**Figure 8. The same as Fig. 7 but using wider soil water content range by adding one more GPR measurement before the rainfall (black dot in Fig. 3a). Excellent performances of the approach are achieved in S2 and S3, while only a little improvement happens in S1 compared to Fig. 7. Apart from the soil water content range, the slope-induced lateral water redistribution is also essential to the proposed approach.**