# Peer review of "Efficient estimation of effective hydraulic properties of stratal undulating surface layer using time-lapse multi-channel GPR"

_Hydrology and Earth System Sciences, 2019_

## Referee Comment (RC1) · Anonymous Referee #1 · 21 May 2019

In this paper, Pan et al. evaluated the performance of time-lapse multi-channel GPR in estimating stratal soil hydraulic parameters at plot scale. Some key factors in the inversion were discussed based on a series of synthetic and field studies. In my opinion, this paper is interesting, generally well written and easy to follow, I recommend its publication after a moderate revision. The following comments may be helpful in improving this manuscript.

1. "structural errors" are mentioned many times in the manuscript. However, an explicit definition is absent. What are included in the structure errors? Are they only related to the geometric error in the underlying interface between the two layers?

[Figure]

2. Page 6 Line 4. The Latin-hypercube was used to generate the initial ensemble. Please provide and justify the initial (prior) statistics for the hydraulic parameters. Also, do you consider the correlation between soil hydraulic parameters? This might have impacts on your inversion results, e.g., Scharnagl 2011 and Man 2016.

3. Please provide some information regarding the computational cost in the inversion since "efficient estimation" is highlighted in the title. For example, how many CPU-hours were needed in 40 iterations in your field case. What about the computational cost in a single model evaluation if a 3D model is considered to cope with the lateral flow?

4. Page 6. The early stopping may cause the overestimation of uncertainties. How do you choose an appropriate iteration number in practical applications? Please clarify.

5. Please shorten the caption of figure 4 since it is rather long.

6. Figure 5 shows significant unresolved biased errors. If I understand correctly, is it possible to alleviate this problem by incorporating geometric stratal errors in the architectures? To be more specific, it seems that all the inversions are based on the same interface (shown in Figure 2). Can you use an interface ensemble instead? This is similar to the treatment of using initial parameter ensemble in your inversion.

References: B Scharnagl, JA Vrugt, H Vereecken, M Herbst. Inverse modelling of in situ soil water dynamics: Investigating the effect of different prior distributions of the soil hydraulic parameters. Hydrology and Earth System Sciences, (2011). 15 (10), 3043-3059. J Man, W Li, L Zeng, L Wu. Data assimilation for unsaturated flow models with restart adaptive probabilistic collocation based Kalman filter. Advances in water resources (2016) 92, 258-270

---

## Referee Comment (RC2) · Anonymous Referee #2 · 1 Jun 2019

This is an interesting paper. Hydraulic properties of the surface layer was estimated using time-lapse multi-channel GPR at plot scale. I recommend its publication after a moderate revision. The following comments may be helpful for improving the manuscript. 1. The term 'effective hydraulic properties' is used in the paper. What's the difference of effective hydraulic properties and hydraulic properties? 2. Page 5 Line15 : For the bottom boundary, a Dirichlet boundary condition is applied with a fixed water pressure of -0.4 m. Page 9 Line12 : The lower Dirichlet boundary is constantly set to the position of the water table inferred from drilling (-1.7 m). Please explain the bottom boundary. Is 'a fixed water pressure of -0.4 m' a good assumption? How will this assumption affect results? 3. Page 14 Line19 : Particularly, the structural information

resulting from the slope-induced lateral water redistribution is essential to the proposed approach. Page 16 Line15 : Application of the demonstrated approach mainly relies on the slope gradient of the undulating structure and the lateral water redistribution. A 2D model is used here to simulate lateral water redistribution. If a 3D model is applied here, the simulated lateral water redistribution may be more or less different. How do you consider this? 4. Page 9 Line6 : For the hydraulic model of the transect, the geometry (16.82 m x 1.3 m) is discretized with a resolution of 0.10 m x 0.05 m (Fig. 3b). Page 13 Line4 : Three two-layer architectures (S1, S2, and S3 in Fig. 6) are employed with a domain of 6.28 m x 2 m and are discretized 5 with a resolution of 0.04 m x 0.02 m. Two different models are used instead of one, please explain.

---

## Author Comment (AC1) · 9 Jul 2019

**Reply to Referee #1:**

**General comments**
**Reply:** We thank the reviewer for the constructive comments and suggestions. We revised the manuscript and thus refer to the revised manuscript.

1. "structural errors" are mentioned many times in the manuscript. However, an explicit definition is absent. What are included in the structure errors? Are they only related to the geometric error in the underlying interface between the two layers?

**Reply:** We understand "structural errors" as non-Gaussian deviations between the model and the measurement data, typically due to simplifications in the representation of the physical processes (more information is given, e.g., in Jaumann and Roth, 2018).

 As listed in the original manuscript (P9 Line 20 – P10 Line 3), the structural errors present in this study mainly include two sources. One is the conceptual error originating from the simplification of 3D flow to 2D, the other one is mainly related to errors in the GPR observations (reflector depth and water storage). These errors are due to the varying accuracy of multi-channel GPR evaluation with different ratios of antenna separation to reflector depth, small-scale soil heterogeneity, and the interpolation of unevenly spaced data.

2. Page 6 Line 4. The Latin-hypercube was used to generate the initial ensemble. Please provide and justify the initial (prior) statistics for the hydraulic parameters. Also, do you consider the correlation between soil hydraulic parameters? This might have impacts on your inversion results, e.g., Scharnagl 2011 and Man 2016.
**Reply:** We are aware of the literature covering the effect of the prior statistics on the estimation of hydraulic parameters. In order not to bias the results with possibly wrong prior assumptions on the hydraulic parameters of the soils, we drew an ensemble of uniformly distributed parameters using the Latin-hypercube algorithm. Thus, the estimated parameters are solely based on the given model and the measurement data.

*References: B Scharnagl, JA Vrugt, H Vereecken, M Herbst. Inverse modelling of in situ soil water dynamics: Investigating the effect of different prior distributions of the soil hydraulic parameters. Hydrology and Earth System Sciences, (2011). 15 (10), 3043-3059.*
*J Man, W Li, L Zeng, L Wu. Data assimilation for unsaturated flow models with restart adaptive probabilistic collocation based Kalman filter. Advances in water resources (2016) 92, 258-270*

3. Please provide some information regarding the computational cost in the inversion since "efficient estimation" is highlighted in the title. For example, how many CPU hours were needed in 40 iterations in your field case. What about the computational cost in a single model evaluation if a 3D model is considered to cope with the lateral flow?
**Reply:** For our field case, the total time to complete the inversions of all 30 ensemble members was about 4x8x20=640 core hours (2 threads per core). The evaluation of the GPR data is comparably efficient, since the undulating surface may require a full Maxwell solver for the simulation of the GPR data increasing the computational cost by at least one order of magnitude.

 A 3D model could deal with the lateral flow, however the corresponding computational cost

multiplies approximately by the number of grid cells in the third dimension. Thus, we decided to proof of our approach using a 2D model. Certainly, going 3D requires scaling the computational power, however it does not require additional concepts to what is presented in this study.

4. Page 6. The early stopping may cause the overestimation of uncertainties. How do you choose an appropriate iteration number in practical applications? Please clarify.
**Reply:** Commonly, the "optimal stopping" criterion (stop when meeting one of the convergence criterions) should be enough for ideal conditions, but overfitting might happen when modelling uncertainties are notable. To avoid overfitting, we used a two-step criterion for the field study. The complete procedure is that the "optimal stopping" criterion is applied with a number of iterations up to 40 at first, then the "early stopping" iteration number is identified after an analysis of the convergence behavior of the parameter distribution. As a rule of thumb, for the "early stopping", we analysed the parameters of the according iteration number (e.g., 5) where the spreading of the cost values becomes stable as shown in the following figure. Generally, the "optimal stopping" criterion should be applied once the structural errors are alleviated in the field study.

[Figure]

We clarify this in the P10 lines 4-10 as "Hence, a two-step procedure is applied in this study. Initially, we used the "optimal stopping" criterion only considering measurement precision and it resulted in a large number of iterations as well as overfitting. After an analysis of the convergence behaviour of the parameter distribution, we identified that overfitting could be avoided by using the "early stopping" criterion and evaluating the output of the 5th iteration. Note that the "optimal stopping" criterion should be applied once the structural errors are alleviated in the field study."

5. Please shorten the caption of figure 4 since it is rather long.
**Reply:** It is rephrased as following.
"Figure 4. Effective hydraulic properties estimated with the proposed approach for the field study (upper panel) and the synthetic study (lower panel). The grey curves in all subplots

represent the estimated hydraulic functions from ensemble members using the "early stopping" criterion. The black curves in (a) and (b) are derived from the median parameters of the best 68 members, while the black curves in (c) and (d) are the same members but using the "optimal stopping" criterion. The red curves in (c) and (d) are derived from the true parameter given in Table 2. The histograms in (a) and (c) (blue histogram, the same in Fig. 6 and Fig. 7) mark the covering range of the mean water content evaluated from the GPR observations."

6. Figure 5 shows significant unresolved biased errors. If I understand correctly, is it possible to alleviate this problem by incorporating geometric stratal errors in the architectures? To be more specific, it seems that all the inversions are based on the same interface (shown in Figure 2). Can you use an interface ensemble instead? This is similar to the treatment of using initial parameter ensemble in your inversion.

**Reply:** The remaining deviations between the model and the measurement data are related to the structural errors which are not restricted to the geometric stratal errors, but also include errors concerning water storage estimation, analysis of the GPR data, and the conversion from 3D to 2D. This can be analysed with a Bayesian total error analysis (BATEA; Kavetski et al., 2002, 2006) by creating ensembles of different models that can represent these structural errors and using this ensemble for parameter estimation. However, this is a major effort that goes well beyond the scope of this study.

Kavetski, D., S. Franks, and G. Kuczera (2002), Confronting input uncertainty in environmental modelling in calibration of watershed models, in Water Sci. Appl. Ser., vol. 6, edited by Q. Y. Duan, et al., pp. 49–68, AGU, Washington, D. C.

Kavetski, D., G. Kuczera, and S. W. Franks (2006), Bayesian analysis of input uncertainty in hydrological modeling: 1. Theory, Water Resour. Res., 42, W03407, doi:10.1029/2005WR004368.

---

## Author Comment (AC2) · 9 Jul 2019

**Reply to Referee #2:**

**General comments**
**Reply:** We would like to thank the reviewer for their evaluation and constructive comments, which definitely helped to improve our paper.

1. The term 'effective hydraulic properties' is used in the paper. What's the difference of effective hydraulic properties and hydraulic properties?
**Reply:** Commonly, we quantify soil hydraulic properties for a small soil sample and implicitly assume homogeneity within the measured volume. However, soil heterogeneity may become non-negligible when the scale of the measurement volume is increased. Particularly, this is required for the analysis of natural sediments. Thus, effective hydraulic properties are used to define the heterogeneous soils. In this study, we focus on the hydraulic properties at plot scale where soil heterogeneity does exist. Therefore, we refer to effective hydraulic properties.

2. Page 5 Line15: For the bottom boundary, a Dirichlet boundary condition is applied with a fixed water pressure of -0.4 m. Page 9 Line12: The lower Dirichlet boundary is constantly set to the position of the water table inferred from drilling (-1.7 m). Please explain the bottom boundary. Is 'a fixed water pressure of -0.4 m' a good assumption? How will this assumption affect results?
**Reply:** Thanks for pointing out this mistake. The two numbers originate from different references. The measured water table inferring from drilling is 1.7 m below the ground surface and the height of the 2D geometry model is 1.3 m. Hence, the water pressure at the bottom is set as -0.4 m (1.3m - 1.7m). To keep consistence, we revise it in Page 9 "The lower Dirichlet boundary is constantly set to a pressure of -0.4 m in consideration to a water table of 1.7 m below ground surface inferred from drilling.".

   Since our experiment was conducted on farmland, we didn't have in-situ monitoring. Thus, we set the bottom boundary condition as a fixed water table. We acknowledge that this assumption can bring some uncertainties in water content distribution in the simulated domain, because the water table may have varied in time and space during the observed period. However, the inversion mainly relies on the information from the upper layer. Besides, we know from a synthetic study that the sensitivity of the parameters of the first layer typically surpasses that of the bottom layer by two orders of magnitudes. Therefore, although the simplification of the bottom boundary condition increases the structural errors, we decided to neglect this source of uncertainty in a first step.

3. Page 14 Line19: Particularly, the structural information resulting from the slope-induced lateral water redistribution is essential to the proposed approach. Page 16 Line15: Application of the demonstrated approach mainly relies on the slope gradient of the undulating structure and the lateral water redistribution. A 2D model is used here to simulate lateral water redistribution. If a 3D model is applied here, the simulated lateral water redistribution may be more or less different. How do you consider this?
**Reply:** The advantage of 3D model is that more lateral water redistribution information can be captured and the conversion error from 3D to 2D can also be avoided. However, from the GPR data of the material interface suggest, that the main slopes contributing to the lateral flow can be well approximated with a 2D model. Thus, in this study, we focus on the general proof of our concept using the 2D model. concerning the expensive computational cost of 3D modelling,

please also note the answers to the comments 3 and 6 of referee 1.

4. Page 9 Line6: For the hydraulic model of the transect, the geometry (16.82 m x 1.3 m) is discretized with a resolution of 0.10 m x 0.05 m (Fig. 3b). Page 13 Line4: Three two-layer architectures (S1, S2, and S3 in Fig. 6) are employed with a domain of 6.28 m x 2 m and are discretized with a resolution of 0.04 m x 0.02 m. Two different models are used instead of one, please explain.

**Reply:** The two models were designed with different goals in mind. The geometry of the field study is given by the field site. In order to minimise the computational cost for the analysis and the parameter estimation, we decided to decrease the resolution of the model.

The synthetic study was designed to analyze the possiblities of the presented approach, once very favourable conditions are met. Thus, we decreased the domain size and increased the resolution of the model.

---

## Author Response (AR2)

Dear editor,

      Thank you very much for pointing out the technical problems. We have corrected them in the marked manuscript.

Kind regards,
Xicai Pan

[revised manuscript text omitted]

*Data availability.* The observational water content and reflector depth data from GPR measurements are available upon request.

*Author contributions.* XP and SJ jointly developed the concept and methodology of the study. XP

5  analysed the measurement and simulated data. SJ set up the simulation approach. KR and JZ contributed with guiding discussions. XP prepared the manuscript with contributions of all authors.

*Competing interests.* The authors declare that they have no conflict of interest.

10  **Acknowledgements**

This work is financially supported by the NSFC (National Natural Science Foundation of China) Project (grant 41771262). We are also grateful to the editor Bob Su and to two anonymous referees, who helped to improve the manuscript significantly.

[revised manuscript text omitted]